# CROSS-MODAL ALIGNMENT AND HUMAN PREFERENCE LEARNING FOR FINE-GRAINED MUSIC-GUIDED IMAGE GENERATION

## ABSTRACT

Mapping temporally evolving musical affect into coherent visual imagery is a challenging instance of cross-modal generation: audio is abstract, layered, and subjective, whereas images are static and concrete. We present MusePainter, a general framework that integrates structured cross-modal alignment with multi-axis preference learning to achieve fine-grained controllability in generative models. MusePainter first extracts structured descriptors capturing structural, stylistic, and affective dimensions of music, which serve as controllable guidance for image synthesis. To handle subjectivity, we introduce a preference optimization scheme that disentangles emotional consistency, semantic alignment, and creative appeal, and optimizes them independently. Experiments on curated benchmarks and user studies demonstrate that MusePainter surpasses strong audio-to-image and audio→text→image baselines in semantic fidelity, stylistic congruence, and affective resonance. While developed for music-to-image, the framework's components—such as interpretable descriptors and multi-axis preference optimization—may also extend to other modalities, offering potential insights for broader controllable cross-modal generation.

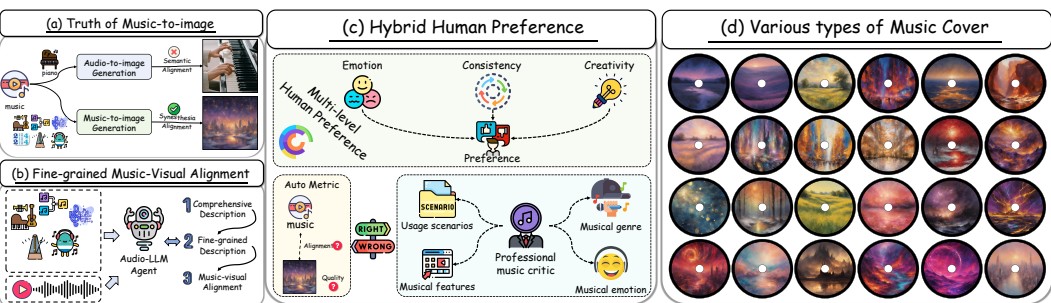

Figure 1: A simple schematic diagram is used to illustrate the tasks and main contributions of this MusePainter.

## 1 INTRODUCTION

Music evokes vivid and subjective imagery, yet translating temporally evolving musical affect into coherent visuals remains an unsolved challenge in cross-modal generation. Unlike environmental sounds that correspond to concrete sources (e.g., traffic or footsteps), music is abstract, layered, and emotionally dynamic. Bridging this gap is scientifically valuable: it requires models to integrate symbolic, stylistic, and affective representations across modalities, pushing beyond the literal semantic mappings that dominate existing multimodal systems.

Recent progress in audio-to-image generation Sung-Bin et al. (2023); Zhao et al. (2022) has demonstrated that cross-modal models can produce visually plausible results for concrete sounds. However, when the input is abstract and time-varying music, these systems fail to capture emotional nuance and stylistic coherence. Two core obstacles stand out. **First, the challenge of cross-modal**

**alignment for abstract affect** (Figure 1a): music evolves continuously across melody, rhythm, and harmony, whereas images are static snapshots. Capturing this mapping demands structured representations that preserve layered musical semantics. **Second, the dilemma of subjectivity in evaluation and control**: different listeners may associate the same melody with vastly different imagery Juslin (2010), making universal ground truth impossible. Objective metrics such as CLIP or FAD cannot fully reflect emotional resonance, while existing pipelines lack mechanisms for fine-grained, preference-aware control.

To address these challenges, we introduce **MusePainter**, a framework that couples interpretable music-theoretic descriptors with human preference learning for fine-grained and controllable music-to-image generation. MusePainter first deconstructs music into its foundational, fine-grained attributes, analyzing elements such as melody, rhythm, harmony, instrumentation, and overall structure to capture the music's emotional texture in a structured manner. This deep musical understanding transcends mere acoustic features to interpret the theoretical underpinnings of the music (Figure 1 (b)). Another key innovation is the Hybrid Human Preference mechanism (Figure 1 (c)). Recognizing the highly personal nature of emotional and aesthetic interpretation, our system is designed to learn from individual preferences. This allows the framework to adapt its visual style to match a user's personal aesthetic and emotional response, thereby generating more meaningful and expressive imagery. This approach directly confronts the challenge of subjectivity that has long limited previous attempts. Our contributions are threefold:

**(1).Problem formalization:** We systematically analyze the semantic gap in music-to-image generation, highlighting abstract emotional alignment and subjective evaluation as two core challenges.

**(2).Framework and method:** We propose MusePainter, a three-stage pipeline that integrates structured music descriptor extraction, cross-modal visual prompt construction, and multi-axis reinforcement learning from human feedback.

**(3).Benchmark and validation:** We curate a descriptive album art benchmark (Figure 1d) and conduct extensive experiments. Results show that MusePainter outperforms strong audio-to-image and audio→text→image baselines in semantic fidelity, stylistic congruence, and affective resonance, validated through both expert-annotated metrics and user studies.

## 2 RELATED WORK

**Audio-to-Image Generation.** Early audio-to-image generation (AIG) methods using direct CNN-based mappings Wan et al. (2019); Lee et al. (2022) were hindered by poor cross-modal semantic alignment Chen & Akata (2021); Mazumder & P (2021); Sun & Liang (2020). Subsequent work improved alignment through specialized strategies Qin et al. (2023); Sung-Bin et al. (2023) and large pre-trained models like CLIP Wu & Bello (2022); Guzhov et al. (2022); Kreuk et al. (2022). Despite these advances, the semantic granularity of audio embeddings remains coarse, limiting their ability to capture fine-grained and affective details. To circumvent this, a recent paradigm shift involves using text as an intermediary to leverage powerful text-to-image (T2I) models Lee et al. (2023); Yariv et al. (2023); Qin et al. (2024); Wang et al. (2023). However, this audio-to-text conversion often discards the nuanced, non-linguistic emotional expressions inherent in music, making the resulting images emotionally shallow—a limitation our work directly addresses.

**Audio-related Cross-Modal Alignment.** Cross-modal alignment for audio has largely mirrored advances in the text–image domain, with contrastive learning frameworks Radford et al. (2021); Wu et al. (2023b) becoming foundational. This led to powerful models like AudioCLIP Guzhov et al. (2022) and CLAP Elizalde et al. (2023), which learn shared embedding spaces for robust audio–text understanding. The state of the art is represented by models such as ImageBind Girdhar et al. (2023), which unifies six modalities into a single space using images as a central anchor. Yet a critical limitation persists: these models are optimized to align concrete, event-based semantics. Their architectures are not designed to capture the abstract, evolving, and emotional contours that characterize music, creating a gap that motivates our work.

**RLHF in T2I Tasks.** Reinforcement Learning from Human Feedback (RLHF) is increasingly replacing automated metrics Heusel et al. (2017); Hessel et al. (2021) for aligning text-to-image (T2I) models with human preferences. Reward models vary in their approach, from providing **holistic scores** to capture overall image quality, as in ImageReward Xu et al. (2023); Wu et al. (2023a), to offering **fine-grained feedback** on specific flaws Liang et al. (2024). Generative models are then

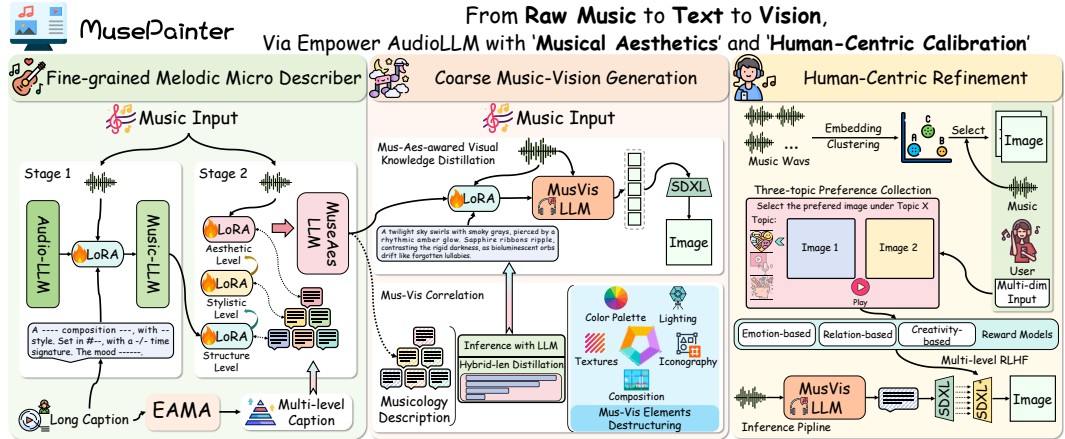

Figure 2: The overall architecture of our proposed MusePainter framework. The framework consists of three main stages. (1) Fine-grained Melodic Micro Describer, (2) Coarse Music-Vision Generation, (3) Human-Centric Refinement.

fine-tuned using these preferences, with algorithms evolving from early RL methods Black et al. (2023); Fan et al. (2023) to the more direct and efficient Direct Preference Optimization (DPO) Wallace et al. (2024); Yang et al. (2024); Clark et al. (2023); Hiranaka et al. (2024). However, nearly all existing methods optimize for a *single fused objective*, whereas in music-guided generation, preferences are inherently multi-dimensional (e.g., emotional consistency vs. creative appeal). This motivates our design of a multi-axis preference optimization scheme that treats each axis independently.

## 3 METHODOLOGY

### 3.1 OVERVIEW

As depicted in Figure 2, our *MusePainter* framework enables a controllable transformation from music to vision through three tightly coupled components. First, the **Fine-grained Melodic Micro Describer (FMMD)** extracts structured descriptors spanning structural, stylistic, and affective dimensions of music. Second, the **Coarse Music-Vision Generation (CMVG)** maps these descriptors into visual prompts and synthesizes an initial image with a diffusion backbone. Finally, the **Human-Centric Refinement (HCR)** aligns outputs with human judgments by disentangling multiple preference axes (emotional, semantic, creative) and optimizing them independently. Together, these modules address the core limitations of prior approaches: inadequate handling of abstract musical semantics and lack of fine-grained, preference-aware controllability.

### 3.2 FINE-GRAINED MELODIC MICRO DESCRIBER

Our approach to fine-grained melodic description unfolds in two stages. First, the Expert-guided Agent-based Music Analysis (EAMA) module converts unstructured, long-form text into a set of structured, high-quality descriptors. Second, these descriptors guide a two-stage, coarse-to-fine finetuning of MuseAes-LLM. This process instills a hierarchical understanding, starting with holistic alignment and progressively refining it with micro-level structural and stylistic details.

**Expert-guided Agent-based Music Analysis (EAMA).** To obtain fine-grained semantic representations from long-form music descriptions, we introduce the **Expert-guided Agent-based Music Analysis (EAMA)** module (Fig. 3). Given a music clip $x$ and its text description $t$, EAMA's goal is to extract a structured set of descriptors $\mathcal{D}(t)$ across six dimensions: instrument, style, key, time signature, tempo, and mood. The process begins with specialized LLM-based agents, each prompted to extract and rephrase content for a single dimension $i$:

$$d^i = \text{Agent}_{\text{LLM}}^{(i)}(t). \tag{1}$$

Figure 3: Framework of Expert-guided Agent-based Music Analysis (EAMA) module, which consists of Hybrid Filter and Multi-level Musicology Description.

To ensure diversity, our *hybrid-length generation strategy* produces candidate descriptions $\mathcal{L}_i = \{d^i_{20}, d^i_{50}, d^i_{70}\}$ at three token lengths (20, 50, 70), corresponding to concise, balanced, and expressive abstraction levels. To mitigate hallucinations and ensure factual grounding, we employ a *hybrid filtering mechanism*. For each dimension $i$, we compute a multi-agent agreement score $s^{\text{agree}}$ and a cross-modal consistency score $s^{\text{cross}}$:

$$s^{\text{agree}}(d^i) = \frac{1}{N} \sum_{n=1}^{N} \text{sim}(d^i, \text{Agent}_n(t)), \quad s^{\text{cross}}(d^i, x) = \text{CLAP}(x, d^i) + \text{ImageBind}(x, d^i). \quad (2)$$

We then retain the descriptor with the highest combined score $s^{\text{agree}} + s^{\text{cross}}$. This yields a final output of structured, multi-granularity descriptors:

$$\mathcal{D}(t) = \{\mathcal{L}_{\text{instr}}, \dots, \mathcal{L}_{\text{mood}}\}. \quad (3)$$

**Two-stage Music-LLM Finetune.** We develop *MuseAes-LLM* by finetuning a Qwen-Audio Chu et al. (2023) base model in two stages. The intuition is that coarse-level finetuning first ensures global semantic alignment, while fine-grained hierarchical learning subsequently injects detailed control over structural, stylistic, and affective aspects.

*Stage 1: Coarse-level Alignment.* We align the model with holistic music descriptions by finetuning on $(x, t)$ pairs, where $x$ is the input audio. The coarse-level objective is:

$$\mathcal{L}_{\text{coarse}} = \mathbb{E}_{(x,t)} \left[ -\log P(t \mid x; \theta_{\text{coarse}}) \right]. \quad (4)$$

*Stage 2: Fine-grained Hierarchical Learning.* Next, we perform descriptor-level tuning. The descriptors are grouped into three semantic levels: Structural ($\{d^{\text{sig}}, d^{\text{tempo}}, d^{\text{key}}\}$), Stylistic ($\{d^{\text{instr}}, d^{\text{style}}\}$), and Aesthetic ($\{d^{\text{mood}}\}$). This is optimized via a hierarchical multi-task loss:

$$\mathcal{L}_{\text{fine}} = \sum_{l \in \{\text{struct}, \text{style}, \text{aesthetic}\}} \lambda_l \cdot \mathbb{E}_{(x, d^l)} \left[ -\log P(d^l \mid x; \theta_{\text{fine}}) \right], \quad (5)$$

where $d^l$ is the concatenated descriptor for level $l$, and $\lambda_l$ is its corresponding weight. To further enhance the model's capacity to interpret musical attributes, we adopt an *instruction-style finetuning strategy*: each descriptor dimension is paired with a tailored prompt (e.g., *"Describe the emotional tone of this music"*), so that the model learns to answer specific evaluative queries in a controlled manner.

### 3.3 COARSE MUSIC-VISION GENERATION (CMVG)

To bridge music and vision, we map our three-tiered music descriptors (structural, stylistic, aesthetic) to five visual dimensions: color palette, lighting, iconography, composition, and textures. We then introduce **MusVis-LLM**, a dedicated model trained to generate structured visual descriptions directly from audio. This module serves as a critical bridge between music understanding and visual synthesis, providing controllable coarse grounding for subsequent refinement.

**Hybrid-Length Visual Description Distillation.** We first construct pseudo-ground-truth visual descriptions for each music clip. A reasoning-focused LLM ($\text{LLM}_{\text{inf}}$) is prompted to infer a textual description for each of the five visual dimensions. To balance conciseness and expressiveness, we employ a *Hybrid-Length Distillation* strategy. For each visual dimension $v$, the LLM generates outputs of three fixed lengths (20, 50, and 70 tokens), corresponding to factual, balanced, and

expressive abstraction levels. Empirically, we find that different lengths capture complementary properties: shorter (20-token) captions reduce hallucination but may omit details, longer (70-token) captions provide rich stylistic cues at the cost of factual precision, while medium-length (50-token) captions strike the best balance (see Fig. 4). As shown in the figure, the 50-token outputs maximize the balanced score while maintaining low hallucination, justifying our choice of 20/50/70 as anchor lengths. The 50-token candidate is then refined by distilling knowledge from the 70-token (teacher) and 20-token (reference) variants, combining detailed stylistic cues with factual grounding:

$$\tilde{y}_v = \text{Distill}(y_v^{50};\ y_v^{70},\ y_v^{20}). \tag{6}$$

This process ensures that the resulting description preserves both accuracy and stylistic richness.

**Mus-Aes-awared Visual Knowledge Distillation.** The distilled outputs for all five dimensions are concatenated into a single structured prompt:

$$\tilde{y} = \text{concat}(\tilde{y}_{\text{color}}, \tilde{y}_{\text{lighting}}, \dots, \tilde{y}_{\text{textures}}). \tag{7}$$

This prompt $\tilde{y}$ acts as a pseudo-label to finetune MusVis-LLM on music-to-text generation, with the training objective:

$$\mathcal{L}_{\text{musvis}} = \mathbb{E}_{(x,\tilde{y})}\left[-\log P(\tilde{y} \mid x; \theta_{\text{musvis}})\right]. \tag{8}$$

During inference, MusVis-LLM generates structured visual descriptions from audio input $x$, which are then passed into a diffusion-based text-to-image model (e.g., SDXL) to synthesize a coarse visual output. This stage provides interpretable and controllable grounding, which is further refined in the Human-Centric Refinement (HCR) module.

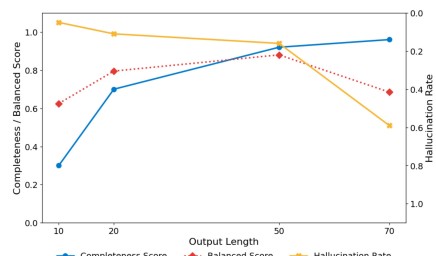

Figure 4: Performance metrics for visual description evaluated at different output lengths. The red dotted line represents the Balanced Score, which is a combined metric of the other two.

### 3.4 HUMAN-CENTRIC REFINEMENT (HCR)

To align image generation with human preferences, we implement a multi-task Reinforcement Learning from Human Feedback (RLHF) pipeline. This process refines the generator independently across three preference axes without fusing reward signals, thereby avoiding reward collapse and enabling interpretable control. The pipeline involves three stages:

**Cluster-based Candidate Selection.** To create informative data for annotation, we first cluster all music clips in the ImageBind embedding space using $k$-means. For each music clip $x$, we then form an annotation triplet $\{z_{\text{orig}}, z^+, z^-\}$. The positive candidate $z^+$ is the image with the highest embedding similarity to $x$ within the same cluster. The negative candidate $z^-$ is randomly sampled from the most distant cluster. This contrastive sampling increases the difficulty of the comparison and enhances the strength of the human preference signal.

**Multi-Topic Preference Annotation.** Human annotators provide feedback on three separate topics: *emotional consistency*, *semantic content alignment*, and *creative aesthetic appeal*. Annotators score each candidate pair sequentially on these topics, yielding three independent preference datasets $\{\mathcal{D}_e, \mathcal{D}_c, \mathcal{D}_{cr}\}$. Unlike most prior RLHF work that collapses feedback into a single scalar objective, we treat these axes separately to preserve their distinct roles and to support fine-grained control.

**Multi-level RLHF.** Using the collected annotations, we train three separate reward models (RMs): *Emo-RM*, *Con-RM*, and *Cre-RM*, parameterized by $\{\theta_e, \theta_c, \theta_{cr}\}$. Each is optimized on its respective preference dataset $\mathcal{D}_k$ using a pairwise ranking loss:

$$\mathcal{L}_{\text{RM}_k} = -\mathbb{E}_{(t,z_i,z_j)\sim\mathcal{D}_k}\left[\log \sigma\left(f_{\theta_k}(t, z_i) - f_{\theta_k}(t, z_j)\right)\right]. \tag{9}$$

We then fine-tune the image generator (e.g., SDXL) by treating each preference as an independent task. For each axis $k$, we define a separate RLHF loss:

$$\mathcal{L}_{\text{RLHF},k} = -\mathbb{E}_{x,z_0}\left[f_{\theta_k}(x, z_0)\right], \quad k \in \{e, c, cr\}. \tag{10}$$

The final training objective combines the original pre-training loss with the axis-specific RLHF loss, weighted by a hyperparameter $\lambda_k$:

$$\mathcal{L}_k = \mathcal{L}_{\text{pre}} + \lambda_k \mathcal{L}_{\text{RLHF},k}, \tag{11}$$

where $\lambda_k$ is tuned via validation experiments. Unlike alternating or fused-reward schemes, our approach directly optimizes one axis at a time, ensuring stability while preserving targeted improvements. Overall, this multi-axis RLHF framework enables controllable refinement of the generator. It provides interpretable gains on each dimension of human preference—emotional, semantic, and creative—while avoiding the instability and loss of diversity that arise from reward fusion.

# 4 EXPERIMENTS

## 4.1 EXPERIMENT SETTINGS

**Datasets.** Our training set consists of approximately 56,000 music-text pairs. This includes 50,000 pairs from the Flux-Music dataset Fei et al. (2024) and 6,000 clips of Chinese-style music. For the latter, we generated detailed descriptions from metadata and summaries, which were then filtered for quality by a multimodal agent. For evaluation, we utilize two standard benchmarks: Music-Caps Agostinelli et al. (2023) and Song-Describer Manco et al. (2023). MusicCaps provides 5,500 10-second audio clips, each with high-quality annotations from professional musicians. The Song-Describer dataset contains 706 licensed, high-fidelity recordings, which allows for assessing model generalization on professionally produced music.

**Implementation Details.** Our framework is implemented in PyTorch with mixed-precision (CUDA 11.6, NVIDIA Apex) on a single server equipped with four NVIDIA A100 GPUs (40 GB each). The full training pipeline—including music encoder fine-tuning, vision synthesis, and RLHF refinement—requires approximately 150 GPU hours. We employ AdamW and PPO optimizers, a cosine learning-rate scheduler with warmup, and fix all random seeds to ensure reproducibility.

**Evaluation Metrics: Metrics for Music-to-Text.** We adopt *Frechet Audio Distance* (**FAD**) Kilgour et al. (2019), *Kullback–Leibler divergence* (**KL**) Kreuk et al., *CLAP score* Elizalde et al. (2023), and *Fréchet Distance* (**FD**) for quantitative evaluation. FAD and FD capture distributional discrepancies between real and generated audio embeddings, KL measures divergence in predicted label distributions, while CLAP score quantifies audio–text alignment via multimodal embedding similarity. **Metrics for Music-to-Image.** We employ **Image–Music Similarity Metric (IMSM)** Chowdhury et al. (2024) to assess alignment between generated music and conditioning images, leveraging CLIP- and CLAP-based cross-modal similarities. We evaluate artistic fidelity using **BAID** Yi et al. (2023), which provides normalized aesthetic scores for synthesized images. In addition, we compute **CLIP** Radford et al. (2021) similarity between image and text features, and **IMAGEBIND** Girdhar et al. (2023) scores for vision-to-text (Vis2Tex) and vision-to-audio (Vis2Aud) correlations, enabling a fine-grained assessment of multimodal alignment. To provide a holistic evaluation, we introduce the **Balanced Expressiveness Score (BES)**, defined as $\text{BES} = 2.0S_i + 1.0S_s - 2.5P_e$, where $S_i$ denotes the *Intuitive Score* (e.g., energy–saturation correlation), $S_s$ the *Stylistic Score* (e.g., aesthetic mappings such as negative energy–brightness), and $P_e$ the *Extreme Penalty* capturing harmful biases. BES synthesizes these factors into a single measure that balances intuitive grounding with stylistic creativity. Higher BES values indicate a model that better achieves both expressive fidelity and robust alignment.

## 4.2 RESULT ANALYSIS FOR MUSIC-TO-TEXT

We evaluate MusePainter's ability in music-to-text generation through quantitative experiments on MusicCaps and Song Describe. We employ two complementary evaluation protocols: (1) *caption–reference similarity*, measured by cosine similarity between generated and ground-truth texts using CLIP (A), LongCLIP (B), and CLAP (C); (2) *audio–caption alignment*, measured by cosine similarity between audio and generated caption embeddings using CLAP. As shown in Table 6, MusePainter achieves competitive results. On MusicCaps, it matches Qwen-audio on A (0.864) and B (0.914), though ACT_BART still leads across most text-similarity metrics. On Song Describe, MusePainter attains the best audio–text alignment with a CLAP score of 0.501, outperforming ACT_BART (0.445) and Qwen-audio (0.447). A paired Student's $t$-test indicates that this improvement over ACT_BART is statistically significant ($p < 0.05$). We further assess caption quality in a downstream text-to-music generation setting (Table 5). MusePainter consistently achieves superior audio quality scores, obtaining the lowest FD (2.084) and FAD (2.988) on Song Describe, as well

Table 1: Objective comparison of music generation models on the MusicCaps and Song Describe benchmarks. Our model, MusePainter (highlighted), is evaluated against other music- and text-conditioned methods. Lower is better (↓) for KL, FD, and FAD; higher is better (↑) for CLAP. The best results are highlighted in pink.

| model | text | music | Musiccaps | | | | Song describe | | | |
|---|---|---|---|---|---|---|---|---|---|---|
| | | | KL↓ | FD↓ | FAD↓ | CLAP↑ | KL↓ | FD↓ | FAD↓ | CLAP↑ |
| ACT_BART | ✘ | ✓ | 0.861 | 2.522 | 7.125 | 0.254 | 1.629 | 2.553 | 3.275 | 0.202 |
| qwen | ✘ | ✓ | 0.904 | 2.638 | 6.902 | 0.234 | 1.677 | 2.305 | 3.372 | 0.223 |
| MusePainter | ✘ | ✓ | 0.868 | 2.172 | 6.522 | 0.208 | 1.612 | 2.084 | 2.988 | 0.224 |
| MusicGEN | ✓ | ✘ | 1.229 | 2.106 | 3.802 | 0.310 | 1.01 | 2.179 | 5.38 | 0.18 |
| Mousai | ✓ | ✘ | 1.592 | 2.867 | 7.530 | 0.23 | 0.742 | - | 8.320 | 0.29 |
| MusicControlNet | ✓ | ✘ | - | - | 10.81 | 0.22 | - | - | - | - |
| JASCO | ✓ | ✘ | 1.78 | - | 6.05 | 0.26 | 1.39 | - | 4.97 | 0.22 |

Table 2: Experimental Results for Music Caption Task. "A", "B" and "C" denote CLIP, LongCLIP and CLAP, respectively. The best results are highlighted in pink.

| model | Musiccaps | | | Song describe | | |
|---|---|---|---|---|---|---|
| | CLIP↑ | LongCLIP↑ | CLAP↑ | CLIP↑ | LongCLIP↑ | CLAP↑ |
| ACT_BART | 0.902 | 0.939 | 0.567 | 0.868 | 0.912 | 0.445 |
| Qwen-audio | 0.864 | 0.914 | 0.461 | 0.857 | 0.904 | 0.447 |
| MusePainter | 0.864 | 0.914 | 0.459 | 0.867 | 0.911 | 0.501 |

as the best FD (2.172) and FAD (6.522) on MusicCaps. It also ranks second-best in KL divergence across both datasets. Overall, these results demonstrate that MusePainter's captions not only achieve strong semantic similarity but are also highly effective for downstream music generation, producing audio closer to ground truth as reflected by distributional metrics.

## 4.3 ANALYSIS OF MUSIC-TO-IMAGE GENERATION TASK

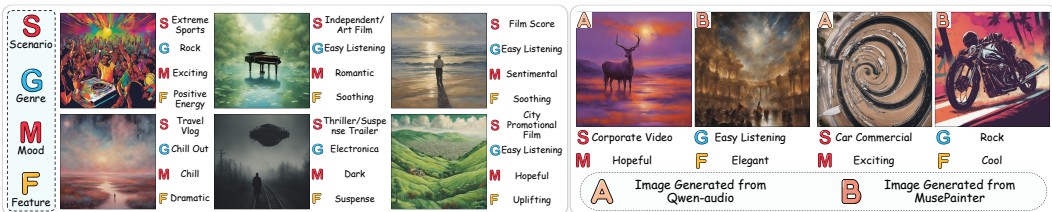

Figure 5: Qualitative Comparison of Music-to-Image Generation. For each audio sample, we show its four annotations (Scenario, Genre, Mood, Feature) alongside the image generated by Muse-Painter, and compare it with the image produced by Qwen-audio via its visual-text pipeline.

**Qualitative Analysis** Figure 10 presents a qualitative comparison of Music-to-Image generation between MusePainter and the Qwen-audio visual-text pipeline. Across the four annotation dimensions—Scenario, Genre, Mood, and Feature—MusePainter demonstrates superior semantic alignment and scene coherence. For instance, given the prompt "Corporate Video + Easy Listening + Hopeful + Soothing," Qwen-audio generated a softly colored stag at sunset. While visually appealing, this image lacks the spatial and professional context of a corporate video. In contrast, MusePainter produced an expansive, warmly lit architectural interior with silhouetted figures, successfully conveying both the corporate setting and the intended optimistic, calming mood. Similarly, for "Car Commercial + Rock + Exciting + Cool," Qwen-audio's output was an abstract vortex of mechanical parts. MusePainter, however, rendered a dynamic, high-speed motorcycle, effectively capturing the excitement and "cool" aesthetic of

a rock-themed automotive ad. These examples highlight MusePainter's strength in translating multidimensional musical inputs into coherent and semantically faithful visual narratives.

**Limitations of General-Purpose Evaluation Metrics** We begin by evaluating several general-purpose cross-modal metrics—IMSM, ImageBind, and BAID—on standard baselines (Table 7). While `Music Des (GEN)` achieves the highest IMSM (14.57 %), and `AudioToken` leads on ImageBind (2.4000) and BAID (5.064), these scores primarily reflect literal, low-level correspondences between audio features or their textual descriptions and the generated images. Such metrics systematically reward models that perform direct semantic mappings (e.g., showing instruments or notation), but they fail to capture the richer *creative* and *emotional* dimensions essential to music-to-image synthesis. Consequently, high-scoring models under these benchmarks may still produce visually uninspired or contextually shallow outputs, highlighting the need for a task-specific evaluation.

Table 3: Comparison of music-to-image generation methods on the IMSM, Imagebind, and BAID metrics. IMSM scores are presented as percentages (%). The best results are highlighted in pink.

| Baseline | IMSM (%) | Imagebind | BAID |
|---|---|---|---|
| Sound2Scene | – | 0.7541 | 4.905 |
| AudioToken | – | 2.4000 | 5.064 |
| Music Des (GEN) | 14.57 | 2.0580 | 4.873 |
| Visual Des (Qwen-ori) | 11.70 | 1.4121 | 4.569 |
| MusePainter-Emo | 11.02 | 1.5819 | 4.457 |
| MusePainter-Con | 11.22 | 1.5408 | 4.498 |
| MusePainter-Cre | 11.24 | 1.4831 | 4.468 |

Table 4: Model performance comparison. The weighted average similarity is calculated to better reflect the priorities of the music-to-image generation task. The weights are assigned with a strong emphasis on semantics: **E**motion (50%), **U**sage Scenarios (30%), **G**enre (15%), and **F**eature (5%). The best results are highlighted in pink.

| Model / Method | CLIP & IMAGEBIND Sim. Score | | | | Avg. | Proposed Metric |
|---|---|---|---|---|---|---|
| | **E** | **F** | **G** | **U** | **Similarity**[*] ↑ | **BES Score** ↑ |
| Sound2Scene-DES | 14.23 | 12.77 | 15.43 | 9.34 | 12.87 | 0.15 |
| AudioToken | 15.22 | 12.98 | 16.41 | 10.25 | 13.80 | 0.15 |
| Music Des | 17.99 | 12.95 | 21.56 | 10.66 | 16.07 | 0.33 |
| Qwen-audio | 19.67 | 13.78 | 19.99 | 10.16 | 16.57 | 0.17 |
| MusePainter-Emo | 20.77 | 13.91 | 18.08 | 12.09 | 17.42 | 0.19 |
| MusePainter-Rel | 20.60 | 14.22 | 17.55 | 11.65 | 17.14 | 0.36 |
| MusePainter-Cre | 20.64 | 14.27 | 17.13 | 12.03 | 17.21 | 0.20 |
| MP w/o CMVG & HCR | 17.72 | 14.49 | 21.55 | 11.60 | 16.30 | 0.18 |
| MP w/o HCR | 19.23 | 13.92 | 18.04 | 10.72 | 16.23 | 0.29 |
| MP-Emo(2k steps) | 19.79 | 13.63 | 18.28 | 10.77 | 16.55 | 0.18 |
| MP-Rel(2k steps) | 20.37 | 13.66 | 17.77 | 10.01 | 16.54 | 0.21 |
| MP-Cre(2k steps) | 20.12 | 13.74 | 16.77 | 10.88 | 16.53 | 0.16 |

**Fine-Grained Semantic Alignment via Expert-Annotated Dimensions** To overcome these limitations, we introduce a specialized framework based on expert annotations along four semantically meaningful dimensions: *Emotion*, *Usage Scenario*, *Genre*, and *Characteristics*. Domain experts labeled reference pairs to establish ground-truth alignments. We then compute per-dimension CLIP & ImageBind similarity scores and aggregate them into a *Weighted Average Similarity* (Emotion 50 %, Usage Scenario 30 %, Genre 15 %, Characteristics 5 %). We also report our proposed *Balanced Expressiveness Score* (BES) for holistic assessment (Table 8). As shown in Table 8, all RLHF-trained variants (`MusePainter-Emo`, `-Rel`, `-Cre`) outperform descriptor-based and other baselines by a substantial margin. Notably, `MusePainter-Emo` attains the highest weighted similarity (17.42), driven by its leading Emotion score (20.77), confirming its effectiveness at capturing affective content. Meanwhile, `MusePainter-Rel` achieves the top BES (0.36), demonstrating a balanced integration of emotional resonance and stylistic fidelity. These quantitative gains validate that our RLHF strategies successfully steer the model toward nuanced

semantic alignment.

**Ablation Study** Our ablation results (Table 8, last five lines) show that removing both CMVG and HCR reduces weighted similarity from 17.42 to 16.30 and BES from 0.19 to 0.18, underscoring CMVG's role in coarse semantic grounding. Ablating only HCR yields a similar drop in similarity (16.23) but a higher BES (0.29), indicating HCR's importance for stylistic refinement. Early RLHF checkpoints (2k steps) plateau around 16.5 similarity and BES 0.16–0.21, demonstrating the necessity of full RLHF training. At 20k steps, the complete models recover and exceed baseline performance, with MusePainter-Emo reaching 17.42 similarity and MusePainter-Rel achieving the top BES of 0.36.

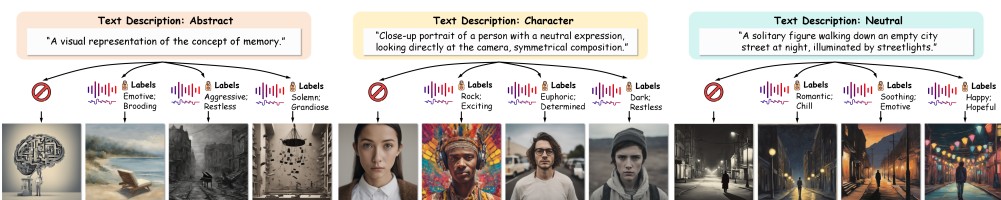

Figure 6: Demonstration of Text- and Music-driven Image Generation. We select three types of text descriptions to vividly showcase MusePainter's capabilities.

**External Experiments** We showcase MusePainter's ability (Figure 11) to incorporate music cues as a fine-grained control signal alongside a fixed text prompt across three scenarios—*Abstract, Character, and Neutral*. In each quartet, the leftmost image is generated by a text-only baseline; the three rightward images illustrate how different audio tracks reshape mood and style while preserving the core concept. For example, for 'character' portrait, a "Dark; Restless" soundtrack yields a brooding, ruin-studded portrait, whereas for 'neutral' portrait, "Happy; Hopeful" transforms "a solitary figure on an empty street" into a lantern-lit festival scene. These results confirm that MusePainter effectively leverages audio attributes to achieve nuanced visual variations beyond text alone.

**Human Analysis** We conducted a user study with 16 participants (8 male, 8 female), each of whom rated outputs from our model (MusePainter) and two baselines (AudioToken, Qwen-audio) on a 1–10 Likert scale. Participants assessed four dimensions—Stylistic Congruence, Rhythmic Correspondence, Image Quality, and Audio-Visual Harmony. As shown in Figure 12, MusePainter attains the highest mean scores in every category, underscoring its superior ability to capture musical style, synchronize visual content with rhythmic structure, generate high-fidelity images, and produce cohesive music-visual pairings. Error bars indicate one standard deviation across all participant ratings.

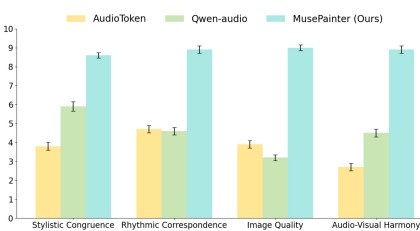

Figure 7: Human Analysis of three methods from four perspectives.

## 5 CONCLUSION

We introduced **MusePainter**, a framework for bridging the semantic and subjective gap in music-to-image synthesis. Our method combines structured music-theoretic descriptors, LLM-driven visual synthesis, and multi-axis reinforcement learning from human feedback (RLHF). Experiments show that MusePainter outperforms strong baselines in semantic fidelity, stylistic coherence, and emotional consistency. These gains are driven by two key innovations: hybrid-length distillation for robust music–visual grounding, and disentangled multi-axis preference optimization for controllable alignment. While developed for music-to-image, the framework's components—interpretable descriptors and multi-axis preference modeling—may extend to other modalities. MusePainter offers interpretable control and lays the foundation for future explorations into dynamic video synesthesia, interactive systems, and broader multimodal datasets.

ETHICS STATEMENT

Our work investigates cross-modal generation by conditioning image synthesis on musical inputs. We believe this research can contribute positively to creative AI, enabling novel forms of artistic expression, advancing understanding of cross-modal alignment, and providing interpretable mechanisms for human-centered generative systems.

Nevertheless, we acknowledge potential negative impacts. Music-to-image generation could be misused for creating misleading or inappropriate visual content, or for reinforcing stereotypes associated with certain musical genres or cultures. Our experiments are conducted exclusively on publicly available benchmark datasets (MusicCaps and Song-Describer), which contain no personally identifiable information. However, biases inherent in these datasets (e.g., cultural bias toward Western music styles) may propagate through our model outputs. We encourage further research on bias detection and mitigation in multimodal generative modeling.

Regarding environmental impact, all experiments were performed on a single NVIDIA A100 GPU (40GB), with training totaling approximately 150 GPU hours. We recognize the importance of efficient model design and responsible use of computational resources to reduce the carbon footprint of large-scale model training.

Overall, we emphasize that our framework should be used only for beneficial and creative purposes. We explicitly discourage applications that may cause harm to individuals, cultures, or society, such as generating deceptive media or infringing upon artistic copyright.

REPRODUCIBILITY STATEMENT

We are committed to ensuring the reproducibility of our work. All datasets used in our experiments are publicly available (MusicCaps Agostinelli et al. (2023), Song-Describer Manco et al. (2023)). We will release our code and trained checkpoints upon acceptance, together with scripts for preprocessing, training, and evaluation.

We describe all necessary implementation details in the main paper and supplementary material, including model architectures (MuseAes-LLM, MusVis-LLM, and SDXL finetuning), optimization settings (learning rate, batch size, optimizer, scheduler), and data preprocessing pipelines. Fixed random seeds were used for all experiments to ensure consistent results across runs.

Our experiments were conducted on a single NVIDIA A100 GPU (40GB). All reported results in tables and figures can be reproduced using the released code and configuration files. We will also provide scripts to regenerate the main figures and evaluation metrics directly from trained checkpoints to facilitate verification and reuse by the community.

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
