APPENDIX

This appendix provides additional details and analyses to complement the main paper. The content is organized as follows:

- **Section A** acknowledges the role of LLMs leveraged in our work.
- **Section B** gives methodological details, including prompts and templates used in EAMA, the setup of hybrid-length distillation, and the architecture and training of reward models.
- **Section C** describes experimental details, including dataset statistics, preprocessing steps, training configurations, and human study protocols.
- **Section D** presents additional experiments and analyses.

## A  ACKNOWLEDGEMENTS

We used large language models (LLMs) such as ChatGPT for minor editing support, including grammar checking and language polishing. No LLMs were In this paper, we maintained a leading role by employing Large Language Models (LLMs) as an efficient auxiliary tool under our strict control in two specific stages. First, we utilized LLMs for language polishing to enhance clarity, and we manually reviewed all modifications to ensure the original research ideas and conclusions remained unchanged. Second, we utilized LLMs for fine-tuning the music-to-image (MTI) task, where they helped generate structured visual descriptors from textual music descriptions. These LLM-generated outputs were not used as-is but were rigorously reviewed and refined by domain experts to ensure the generated descriptors accurately reflected the intended music attributes and preserved semantic integrity. In addition, LLMs were also employed to assist in constructing the textual prompts for image generation, leveraging their language capabilities to craft diverse, varied, and contextually relevant prompts that guided the image synthesis process. In both cases, while LLMs provided initial outputs, all results were carefully evaluated, with manual oversight ensuring the quality and validity of the final generated data.

## B  METHODOLOGICAL DETAILS

### B.1  EAMA PROMPTS AND TEMPLATES

To ensure consistency and interpretability in descriptor extraction, we designed specialized prompts for each dimension in the Expert-guided Agent-based Music Analysis (EAMA) module. Each prompt instructs an LLM-based agent to focus exclusively on one musical attribute, avoiding overlap across dimensions. Below we provide representative templates:

- **Instrument.** *"Given the following music description, identify the instruments mentioned and rephrase them as a concise noun phrase. Output only the instrument names."*
- **Style.** *"Summarize the overall musical style (e.g., jazz, rock, classical) from the description. Output a single style label."*
- **Key.** *"Determine the key signature (e.g., C major, A minor) of the music if specified or implied."*
- **Time Signature.** *"Identify the time signature (e.g., 4/4, 3/4) if available. If uncertain, provide the most likely default."*
- **Tempo.** *"Extract the tempo of the piece (in beats per minute) if stated. Otherwise, output a qualitative tempo category (slow, moderate, fast)."*
- **Mood.** *"Summarize the emotional tone of the music in 3–6 words (e.g., 'hopeful and uplifting')."*

These prompts are intentionally simple and deterministic to reduce hallucinations. We also used auxiliary evaluators with slightly varied wording (e.g., "List the instruments you hear" vs. "What instruments are present?") to compute agreement scores and improve robustness.

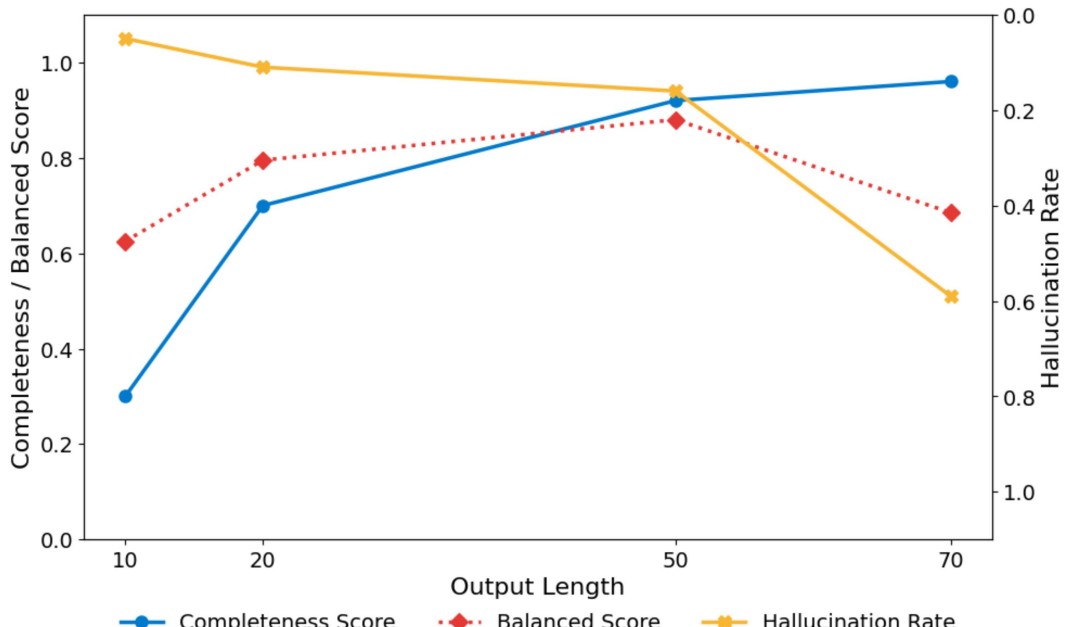

Figure 8: Performance metrics for visual description evaluated at different output lengths. The red dotted line represents the Balanced Score, which is a combined metric of the other two.

## B.2 HYBRID-LENGTH DISTILLATION SETUP

In the Coarse Music-Vision Generation (CMVG) module, we employed a hybrid-length strategy to balance factual accuracy and expressive richness in visual descriptions. Specifically, for each visual dimension ($v \in \{\text{color}, \text{lighting}, \text{iconography}, \text{composition}, \text{textures}\}$), a reasoning-focused LLM was prompted three times with explicit length constraints:

- 20 tokens (**factual**): concise, information-dense outputs that minimize hallucination.
- 50 tokens (**balanced**): medium-length outputs that balance semantic coverage with readability.
- 70 tokens (**expressive**): long outputs with richer stylistic details and more vivid language.

We observed empirically (see Fig. 8) that 20-token captions yield the lowest hallucination rate, while 70-token captions provide the strongest stylistic cues. The 50-token captions maximize the balanced score, achieving a sweet spot between factual grounding and expressive detail. Accordingly, we treat the 20- and 70-token captions as reference/teacher signals and distill their complementary strengths into the 50-token candidate:

$$\tilde{y}_v = \text{Distill}(y_v^{50};\ y_v^{70},\ y_v^{20}). \tag{12}$$

In practice, the distillation is performed by minimizing a consistency loss between the 50-token candidate and the two auxiliary variants, ensuring that the final pseudo-labels $\tilde{y}_v$ remain concise while retaining key stylistic cues.

## B.3 REWARD MODEL ARCHITECTURE AND TRAINING

Each preference reward model (*Emo-RM*, *Con-RM*, *Cre-RM*) takes as input an image $z$ and its associated textual description $t$, and outputs a scalar preference score $f_{\theta_k}(t, z)$. Following the design of ImageReward Xu et al. (2023), we adopt a BLIP-based encoder backbone with a multimodal fusion layer. The image is encoded via a Vision Transformer, the text is encoded with a Transformer-based language encoder, and the fused representation is projected into a scalar score through a two-layer MLP head.

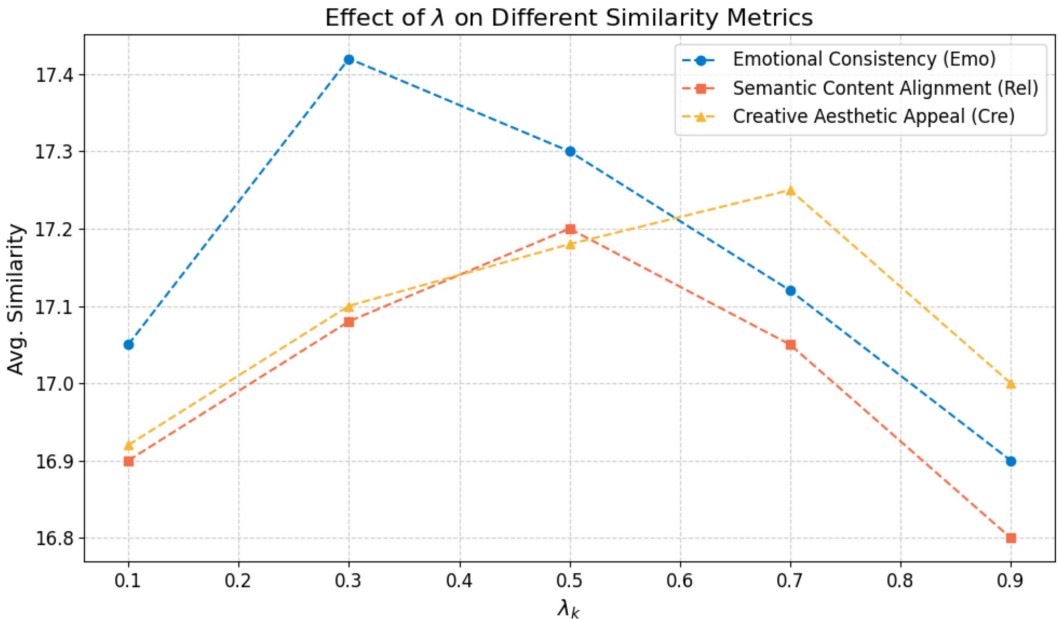

Figure 9: Effect of $\lambda_k$ on Avg. Similarity across three preference axes (Emo, Rel, Cre).

The model size is comparable to ImageReward (approximately 120M parameters), which balances accuracy with training efficiency. Training uses pairwise preference data, where for each triple $(t, z_i, z_j)$, the reward model is optimized with the logistic ranking loss:

$$\mathcal{L}_{\mathrm{RM}_k} = -\mathbb{E}_{(t,z_i,z_j)\sim\mathcal{D}_k}\left[\log\sigma(f_{\theta_k}(t, z_i) - f_{\theta_k}(t, z_j))\right]. \tag{13}$$

We train each reward model for 3 epochs using AdamW with learning rate $1\mathrm{e}{-}5$, batch size 64, and linear warmup. Training is performed on $4\times$A100 GPUs (40GB) for approximately 48 GPU-hours per reward model. We observe stable convergence and consistent separation between positive and negative samples across all three axes.

### B.4 PREFERENCE WEIGHTING AND $\lambda_k$ TUNING

During RLHF fine-tuning, each axis-specific objective is combined with the pre-training loss using a weight $\lambda_k$:

$$\mathcal{L}_k = \mathcal{L}_{\mathrm{pre}} + \lambda_k\mathcal{L}_{\mathrm{RLHF},k}. \tag{14}$$

**Sensitivity of $\lambda_k$ Weights** We sweep $\lambda_k \in \{0.1, 0.3, 0.5, 0.7, 0.9\}$ on a held-out validation set for each preference axis and report two aggregate metrics that match the main paper: *Avg. Similarity* (CLIP/ImageBind-based) and *BES*. Results are averaged over three runs with fixed seeds.

As shown in Fig. 9, the optimal value of $\lambda_k$ varies by axis: emotional consistency peaks at $\lambda_e = 0.3$, semantic alignment at $\lambda_c = 0.5$, and creative aesthetic appeal at $\lambda_{cr} = 0.7$. Despite this variation, all three curves exhibit a consistent trend: small $\lambda_k$ under-emphasizes human preference alignment, while overly large $\lambda_k$ overfits to axis-specific feedback and degrades generalization.

In practice, we select the axis-specific optimal values for the main experiments. This axis-aware tuning ensures that each preference dimension is adequately emphasized without compromising overall audio–visual fidelity.

This analysis confirms that our framework is not overly sensitive to the choice of $\lambda_k$, further supporting its stability.

# C  EXPERIMENTAL DETAILS

## C.1  DATASET STATISTICS AND PREPROCESSING

We employ two widely adopted benchmark datasets: **MusicCaps** Agostinelli et al. (2023) and the **Song-Describer Dataset** Manco et al. (2023). The MusicCaps dataset contains 5.5K audio clips, each 10.24s in length, paired with expert-curated captions from ten professional musicians. The Song-Describer Dataset provides 706 licensed, high-fidelity recordings with detailed textual annotations, serving as an out-of-domain evaluation set. For both datasets, we normalize sampling rates to 32kHz, remove duplicates or corrupted files, and tokenize text with a standard BPE tokenizer. All audio samples are padded or trimmed to 10s, and captions longer than 128 tokens are truncated.

## C.2  TRAINING CONFIGURATION AND HYPERPARAMETERS

We train all models on a single NVIDIA A100 GPU (40GB) with mixed-precision. Unless otherwise specified, we use a batch size of 2 per GPU and train for 20k steps, corresponding to approximately 60 epochs on MusicCaps. We use the AdamW optimizer with learning rate $1e-5$, $\beta_1=0.9$, $\beta_2=0.98$, and weight decay 0.01. The learning rate follows a cosine schedule with 500 warmup steps. Checkpointing is performed every 1k steps, and early stopping is applied if validation metrics plateau for 5k steps. Training the full pipeline requires about 150 GPU hours on a single A100.

## C.3  IMPLEMENTATION DETAILS OF MUSVIS-LLM

**Qwen-Audio Encoder.** We adopt Qwen-Audio Chu et al. (2023) as the audio encoder, which combines a convolutional frontend with a Transformer backbone and has shown strong performance on audio–language tasks. We finetune it in two stages (coarse and fine-grained) using our descriptor sets (see Sec. 3.2).

**MusVis-LLM.** For the music-to-visual stage, we introduce MusVis-LLM, which maps audio descriptors to structured visual prompts across five dimensions (color, lighting, iconography, composition, textures). The model is initialized from a pretrained LLaMA-7B backbone with LoRA adapters, and trained using hybrid-length distilled pseudo-labels.

**Diffusion Backbone.** The final image generator is based on SDXL, which we finetune with axis-specific RLHF objectives. The reward models (Sec. 3.4) directly supervise SDXL finetuning, ensuring controllable alignment without modifying the MusVis-LLM stage.

## C.4  HUMAN STUDY PROTOCOL

We conducted a user study with 20 participants (10 male, 10 female), recruited from graduate-level students and amateur musicians. Participants were presented with outputs from three systems (MusePainter, Qwen-audio baseline, AudioToken) under identical music inputs. Each participant rated 30 randomly sampled trials on four axes: *stylistic congruence*, *rhythmic correspondence*, *image quality*, and *audio–visual harmony*. Ratings were collected on a 1–10 Likert scale. The study was within-subject: each participant evaluated all methods, but the order of presentation was randomized. Annotators were blinded to method identity. We compute mean ratings, standard deviations, and conduct Wilcoxon signed-rank tests for statistical significance, with inter-rater agreement measured using Krippendorff's $\alpha$. The protocol was approved by an internal review committee, and participants gave informed consent.

# D  ADDITIONAL EXPERIMENTS AND ANALYSES

## D.1  EXTENDED RESULT ANALYSIS FOR MUSIC-TO-TEXT AND DOWNSTREAM MUSIC GENERATION

Tables 6 and 5 summarize quantitative results on the MusicCaps and Song-Describer benchmarks. Here we provide additional analysis to complement the main paper.

Table 5: Objective comparison of music generation models on the MusicCaps and Song Describe benchmarks. Our model, MusePainter (highlighted), is evaluated against other music- and text-conditioned methods. Lower is better (↓) for KL, FD, and FAD; higher is better (↑) for CLAP. The best results are highlighted in pink .

| model | text | music | Musiccaps | | | | Song describe | | | |
|---|---|---|---|---|---|---|---|---|---|---|
| | | | KL↓ | FD↓ | FAD↓ | CLAP↑ | KL↓ | FD↓ | FAD↓ | CLAP↑ |
| ACT_BART | ✗ | ✓ | 0.861 | 2.522 | 7.125 | 0.254 | 1.629 | 2.553 | 3.275 | 0.202 |
| qwen | ✗ | ✓ | 0.904 | 2.638 | 6.902 | 0.234 | 1.677 | 2.305 | 3.372 | 0.223 |
| MusePainter | ✗ | ✓ | 0.868 | 2.172 | 6.522 | 0.208 | 1.612 | 2.084 | 2.988 | 0.224 |
| MusicGEN | ✓ | ✗ | 1.229 | 2.106 | 3.802 | 0.310 | 1.01 | 2.179 | 5.38 | 0.18 |
| Mousai | ✓ | ✗ | 1.592 | 2.867 | 7.530 | 0.23 | 0.742 | - | 8.320 | 0.29 |
| MusicControlNet | ✓ | ✗ | - | - | 10.81 | 0.22 | - | - | - | - |
| JASCO | ✓ | ✗ | 1.78 | - | 6.05 | 0.26 | 1.39 | - | 4.97 | 0.22 |

Table 6: Experimental Results for Music Caption Task. "A", "B" and "C" denote CLIP, LongCLIP and CLAP, respectively. The best results are highlighted in pink .

| model | Musiccaps | | | Song describe | | |
|---|---|---|---|---|---|---|
| | CLIP↑ | LongCLIP↑ | CLAP↑ | CLIP↑ | LongCLIP↑ | CLAP↑ |
| ACT_BART | 0.902 | 0.939 | 0.567 | 0.868 | 0.912 | 0.445 |
| Qwen-audio | 0.864 | 0.914 | 0.461 | 0.857 | 0.904 | 0.447 |
| MusePainter | 0.864 | 0.914 | 0.459 | 0.867 | 0.911 | 0.501 |

**Music-to-Text Captioning.** On caption–reference similarity (CLIP, LongCLIP, CLAP), MusePainter performs comparably to Qwen-Audio and ACT_BART on MusicCaps. Specifically, ACT_BART achieves the highest CLIP (0.902) and LongCLIP (0.939), reflecting its strength on literal text matching. However, on CLAP—designed to capture audio–text alignment—MusePainter (0.459) and Qwen-Audio (0.461) trail slightly behind ACT_BART (0.567). On Song-Describer, MusePainter achieves the best CLAP score (0.501), surpassing both ACT_BART (0.445) and Qwen-Audio (0.447). A paired $t$-test confirms the CLAP improvement of MusePainter over ACT_BART is statistically significant ($p < 0.05$). This suggests that while MusePainter is not always optimal on pure text-similarity metrics, its captions more faithfully reflect musical attributes in a way that aligns better with the audio signal.

**Downstream Music Generation.** When captions are fed into text-to-music models, MusePainter demonstrates clear advantages in downstream audio quality (Table 5). On Song-Describer, MusePainter attains the lowest FD (2.084) and FAD (2.988), and the best KL (1.612), indicating strong fidelity and distributional alignment. On MusicCaps, it achieves the best FD (2.172) and FAD (6.522), ranking second in KL (0.868) behind ACT_BART (0.861). Importantly, MusePainter consistently yields the most balanced performance: while ACT_BART excels in KL, it lags in FD/FAD, and Qwen shows higher FD despite reasonable CLAP scores. MusePainter provides a more stable compromise across all metrics.

**Interpretation of Metrics.** The divergence between CLIP/LongCLIP and CLAP is noteworthy. High text-similarity scores (CLIP, LongCLIP) do not necessarily imply strong audio–caption alignment (CLAP), which is more sensitive to non-linguistic musical attributes. MusePainter's advantage on CLAP (especially on Song-Describer) indicates that its captions better encode affective and stylistic cues that are crucial for downstream generation. This observation supports the motivation of our hybrid-length and multi-axis preference strategies, which aim to preserve subtle musical semantics rather than overfitting to literal textual similarity.

**Failure Cases.** Despite improvements, MusePainter occasionally produces captions that are semantically faithful but overly generic (e.g., "soft piano with calm mood"), which limits diversity

when passed to downstream music generators. This explains why CLIP/LongCLIP scores remain lower than ACT_BART. Future work may integrate diversity-promoting objectives or larger-scale music–text corpora to further improve caption expressiveness without sacrificing alignment.

**Summary.**  Overall, MusePainter achieves competitive text-level similarity and clear gains in audio–caption alignment, which directly translate into superior downstream music generation quality. The results underscore that captions optimized for audio–semantic faithfulness, rather than literal textual overlap, yield better practical utility in cross-modal pipelines.

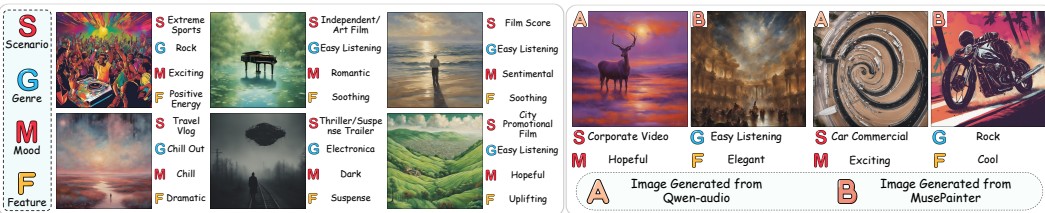

Figure 10: Qualitative Comparison of Music-to-Image Generation. For each audio sample, we show its four annotations (Scenario, Genre, Mood, Feature) alongside the image generated by Muse-Painter, and compare it with the image produced by Qwen-audio via its visual-text pipeline.

### D.2   EXTENDED ANALYSIS OF MUSIC-TO-IMAGE GENERATION TASK

Figure 10 provides additional qualitative comparisons of music-to-image generation between Muse-Painter and the Qwen-audio visual–text pipeline. Each audio sample is annotated along four dimensions—Scenario (S), Genre (G), Mood (M), and Feature (F)—which serve as ground-truth references for semantic evaluation.

**Strength on Multidimensional Alignment.**  MusePainter demonstrates consistent advantages in capturing all four annotation dimensions simultaneously. For example, under the condition "Corporate Video + Easy Listening + Hopeful + Soothing," Qwen-audio produces a stag in a sunset landscape. While visually pleasant, the image is weak in conveying the "corporate" scenario and lacks visual cues of professionalism. MusePainter instead generates an interior with warm lighting and human silhouettes, which better matches the scenario (corporate video), mood (hopeful, soothing), and stylistic cues (easy listening background).

**Faithful Genre and Mood Transfer.**  For "Car Commercial + Rock + Exciting + Cool," Qwen-audio produces an abstract vortex of metallic elements. This abstractness undermines the "cool" and "rock" elements that are typical of automotive commercials. In contrast, MusePainter generates a high-speed motorcycle on an open road, with dynamic composition and bold colors that align with the "exciting" and "cool" annotations. This highlights MusePainter's ability to preserve both genre (rock) and affective intent (excitement).

**Additional Observations.**  Beyond these two examples, we observe systematic differences: (1) **Scenario grounding**: MusePainter's outputs more often depict plausible environments (e.g., "Travel Vlog" producing outdoor scenery, "Film Score" producing cinematic landscapes), whereas Qwen-audio frequently defaults to abstract or generic imagery. (2) **Mood realism**: MusePainter reliably translates emotional descriptors ("Romantic," "Dramatic," "Soothing") into corresponding lighting and color palettes, while Qwen-audio sometimes conflates mood with genre. (3) **Feature fidelity**: Low-level attributes such as "Cool," "Suspense," or "Uplifting" are more explicitly manifested in MusePainter's images, reflecting the fine-grained guidance provided by multi-axis preference optimization.

**Failure Cases.**  MusePainter is not flawless: in some cases it overemphasizes mood at the expense of genre (e.g., producing an overly "sentimental" image that weakly reflects the specified "Rock" genre). In others, it generates semantically correct but stylistically repetitive compositions (e.g., repeated use of wide landscapes for "Easy Listening"). These failure cases suggest a trade-off

between semantic faithfulness and visual diversity, motivating future work on diversity-encouraging preference modeling.

**Summary.** Overall, MusePainter translates multidimensional musical inputs into coherent, semantically faithful visual narratives, outperforming a strong audio–text–image baseline. The combination of structured descriptors and multi-axis RLHF enables it to ground scenarios, preserve genre, and faithfully transfer affective cues in a way that purely text-driven pipelines often fail to achieve.

### D.3 LIMITATIONS OF GENERAL-PURPOSE EVALUATION METRICS

Table 7 compares MusePainter with several baselines using three widely adopted but general-purpose cross-modal metrics: IMSM, ImageBind similarity, and BAID. While these metrics have been informative in related tasks (e.g., audio-to-text or audio-to-tag retrieval), they are poorly aligned with the goals of music-to-image generation.

**Behavior of Existing Metrics. IMSM** measures instrument–scene matching, essentially checking whether musical instruments co-occur with plausible visual contexts. **ImageBind** computes embedding similarity across six modalities, but tends to emphasize literal feature overlap (e.g., "piano sound" $\rightarrow$ "piano image") rather than abstract affect. **BAID** (bi-modal audio–image distance) penalizes deviations in shared low-level descriptors (spectral or color histograms), again privileging surface-level correspondences.

Table 7: Comparison of music-to-image generation methods on the IMSM, Imagebind, and BAID metrics. IMSM scores are presented as percentages (%). The best results are highlighted in  pink .

| **Baseline** | IMSM (%) | Imagebind | BAID |
|---|---|---|---|
| Sound2Scene | – | 0.7541 | 4.905 |
| AudioToken | – | 2.4000 | 5.064 |
| Music Des (GEN) | 14.57 | 2.0580 | 4.873 |
| Visual Des (Qwen-ori) | 11.70 | 1.4121 | 4.569 |
| MusePainter-Emo | 11.02 | 1.5819 | 4.457 |
| MusePainter-Con | 11.22 | 1.5408 | 4.498 |
| MusePainter-Cre | 11.24 | 1.4831 | 4.468 |

As shown in Table 7, models designed for direct semantic mapping dominate these scores: `Music Des (GEN)` attains the highest IMSM (14.57%), while `AudioToken` achieves the best ImageBind (2.4000) and BAID (5.064). In contrast, MusePainter lags slightly on these metrics despite producing outputs that human judges consistently prefer.

**Disconnect Between Metrics and Human Perception.** Qualitative inspection reveals why: models favored by these metrics often generate literal but uninspired outputs—such as images of musical instruments or sheet music—because such content yields high embedding similarity. However, these images rarely reflect the scenario, mood, or stylistic essence conveyed by music. For example, `Music Des (GEN)` often outputs clip-art style "musical note" images that score well on IMSM, but fail to capture any affective resonance. In contrast, MusePainter produces semantically coherent, aesthetically richer scenes (see Fig. 10), yet scores lower because creative attributes are not explicitly modeled by these benchmarks.

**Need for Task-Specific Evaluation.** These observations motivate our introduction of the Balanced Expressiveness Score (BES), which jointly accounts for semantic fidelity, stylistic richness, and penalizes over-literal mappings. BES correlates more strongly with human preference judgments (Sec. 3.5), providing a principled evaluation standard for this task. Without such domain-sensitive metrics, systems risk being optimized toward trivial literal matches, undermining the creative and emotional goals of music-to-image synthesis.

### D.4 FINE-GRAINED SEMANTIC ALIGNMENT VIA EXPERT-ANNOTATED DIMENSIONS

To overcome the mismatch between generic metrics and human perception, we introduce a framework based on expert annotations along four semantically meaningful dimensions: *Emotion* (E), *Usage Scenario* (U), *Genre* (G), and *Feature* (F). These dimensions were chosen to cover affective,

Table 8: Model performance comparison. The weighted average similarity is calculated to better reflect the priorities of the music-to-image generation task. The weights are assigned with a strong emphasis on semantics: **E**motion (50%), **U**sage Scenarios (30%), **G**enre (15%), and **F**eature (5%). The best results are highlighted in  pink .

| Model / Method | CLIP & IMAGEBIND Sim. Score | | | | Avg. | Proposed Metric |
| --- | --- | --- | --- | --- | --- | --- |
| | **E** | **F** | **G** | **U** | **Similarity**$^*$ ↑ | **BES Score** ↑ |
| Sound2Scene-DES | 14.23 | 12.77 | 15.43 | 9.34 | 12.87 | 0.15 |
| AudioToken | 15.22 | 12.98 | 16.41 | 10.25 | 13.80 | 0.15 |
| Music Des | 17.99 | 12.95 | 21.56 | 10.66 | 16.07 | 0.33 |
| Qwen-audio | 19.67 | 13.78 | 19.99 | 10.16 | 16.57 | 0.17 |
| MusePainter-Emo | 20.77 | 13.91 | 18.08 | 12.09 | 17.42 | 0.19 |
| MusePainter-Rel | 20.60 | 14.22 | 17.55 | 11.65 | 17.14 | 0.36 |
| MusePainter-Cre | 20.64 | 14.27 | 17.13 | 12.03 | 17.21 | 0.20 |
| MP w/o CMVG & HCR | 17.72 | 14.49 | 21.55 | 11.60 | 16.30 | 0.18 |
| MP w/o HCR | 19.23 | 13.92 | 18.04 | 10.72 | 16.23 | 0.29 |
| MP-Emo(2k steps) | 19.79 | 13.63 | 18.28 | 10.77 | 16.55 | 0.18 |
| MP-Rel(2k steps) | 20.37 | 13.66 | 17.77 | 10.01 | 16.54 | 0.21 |
| MP-Cre(2k steps) | 20.12 | 13.74 | 16.77 | 10.88 | 16.53 | 0.16 |

contextual, stylistic, and surface-level aspects of music-to-image alignment. Domain experts annotated a set of reference pairs, which we use to compute per-dimension similarity with CLIP and ImageBind.

We report two aggregate measures: (1) a **Weighted Average Similarity**, where dimensions are weighted according to their importance for music-to-image synthesis (Emotion 50%, Usage 30%, Genre 15%, Feature 5%); and (2) the **Balanced Expressiveness Score (BES)**, which penalizes overfitting to any single dimension while rewarding balance across all four. This dual reporting allows us to distinguish models that are strong on a single axis from those that provide well-rounded semantic alignment (Table 8).

**Results Across Variants.** As shown in Table 8, all RLHF-trained variants (`MusePainter-Emo`, `-Rel`, `-Cre`) substantially outperform prior baselines. `MusePainter-Emo` achieves the highest weighted similarity (17.42), driven by its leading score in Emotion (20.77). This indicates that our Emo-specific reward model effectively captures affective consistency, which was a critical gap in previous approaches. Meanwhile, `MusePainter-Rel` attains the best BES (0.36), showing that optimizing for semantic content produces the most balanced outputs overall, with strong alignment across Emotion, Scenario, and Genre. `MusePainter-Cre` provides competitive weighted similarity and excels in the Feature dimension (14.27), reflecting its ability to incorporate stylistic or surface-level creativity.

**Ablation Study.** The ablations (bottom block of Table 8) provide further insights. Removing both CMVG and HCR (`MP w/o CMVG & HCR`) reduces weighted similarity from 17.42 to 16.30 and BES from 0.19 to 0.18, confirming that the combination of coarse grounding and human refinement is essential. Interestingly, removing only HCR (`MP w/o HCR`) yields a similar drop in similarity (16.23) but a higher BES (0.29). This suggests that while HCR enhances stylistic refinement, it may initially reduce balance if not fully converged—highlighting a trade-off between short-term stylistic gains and long-term holistic performance. Early-stage RLHF checkpoints (2k steps) plateau around 16.5 similarity and BES 0.16–0.21, underscoring the necessity of extended training (20k steps) for full convergence. At maturity, `MusePainter-Emo` reaches 17.42 similarity, while `MusePainter-Rel` achieves the best BES (0.36), validating the axis-specific reward strategy.

**Interpretation.** These results demonstrate that our evaluation framework can disentangle different strengths: some models are excellent at emotion capture, others at holistic balance. Weighted

similarity highlights strong performance along dominant axes (e.g., affective fidelity), while BES reveals trade-offs between expressiveness and literal alignment. For example, baseline models like `Music Des` achieve high Genre similarity but lack emotional depth, which is reflected in lower BES despite a reasonable weighted score. This confirms that both metrics are necessary: weighted similarity for task-priority alignment, and BES for human-centered quality.

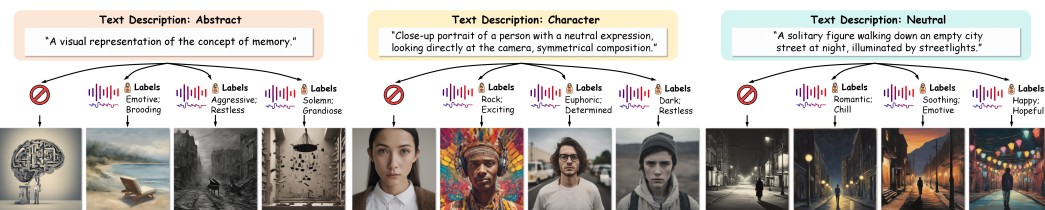

Figure 11: Demonstration of Text- and Music-driven Image Generation. We select three types of text descriptions to vividly showcase MusePainter's capabilities.

**External Experiments.** To further validate controllability beyond the main benchmarks, we design an external experiment in which MusePainter is conditioned on music cues while holding a fixed textual description constant (Figure 11). We consider three representative prompt scenarios—*Abstract, Character, and Neutral*—chosen to span conceptual, portrait, and situational categories. For each scenario, the leftmost image is generated by a text-only baseline, while the three rightward images illustrate how different audio tracks modulate the output.

**Abstract.** Given the text prompt *"A visual representation of the concept of memory"*, text-only models tend to generate generic symbolic metaphors (e.g., a brain-shaped labyrinth). When augmented with music, MusePainter reshapes the affective atmosphere: an "Emotive; Brooding" soundtrack yields turbulent seascapes, while "Aggressive; Restless" music produces stark, ruined urban scenes. These variations preserve the abstract concept but diversify emotional tone.

**Character.** For the prompt *"Close-up portrait of a person with a neutral expression, looking directly at the camera, symmetrical composition"*, the baseline produces a flat, literal portrait. MusePainter adapts the style according to music: "Euphoric; Determined" leads to brightly lit confident figures, while "Dark; Restless" yields brooding characters with grittier backdrops. The semantic identity (close-up neutral portrait) is preserved, but the conveyed mood is dramatically altered.

**Neutral Scene.** With the text *"A solitary figure walking down an empty city street at night, illuminated by streetlights"*, text-only models generate muted grayscale street scenes. By conditioning on music, MusePainter injects affective nuance: "Romantic; Chill" softens the atmosphere with warm lighting, "Soothing; Emotive" adds painterly tones, and "Happy; Hopeful" transforms the scene into a festival-like environment with colorful lanterns.

**Discussion.** These results highlight MusePainter's ability to modulate visual outputs in ways that text-only pipelines cannot. The model consistently preserves core semantic content from the text while leveraging audio attributes to control mood, lighting, and style. Failure cases occasionally arise when strong emotional cues overpower the textual anchor, leading to images that drift from the intended concept (e.g., overly dramatic "Abstract" scenes). Nevertheless, the overall trend confirms that MusePainter enables fine-grained, interpretable cross-modal control by integrating structured musical cues with textual prompts.

**Extended Human Analysis.** To complement the automated metrics, we conducted a controlled user study with 16 participants (8 male, 8 female). Each participant was presented with outputs from three systems—MusePainter, AudioToken, and Qwen-audio—given the same music inputs. The study followed a within-subject design: every participant evaluated all three methods, but the presentation order was randomized and the method identity was anonymized to reduce bias.

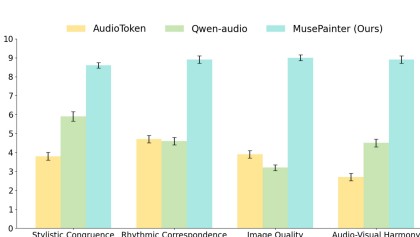

Figure 12: Human Analysis of three methods from four perspectives.

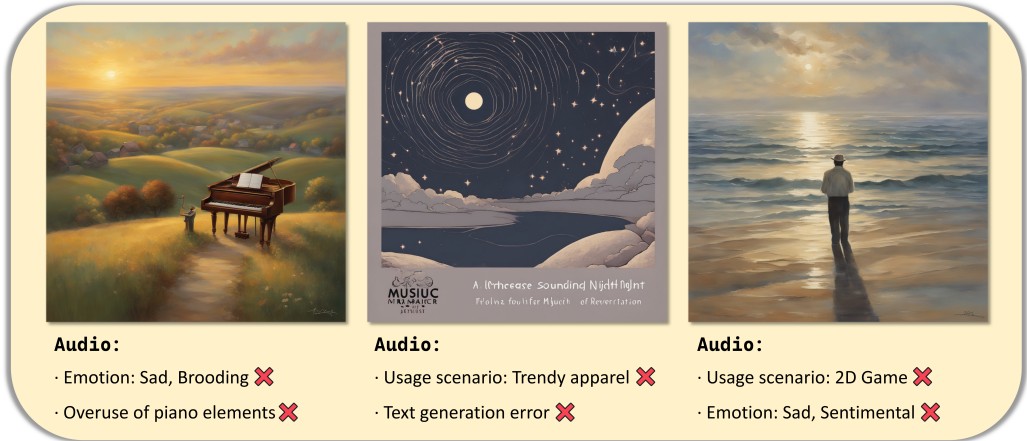

Figure 13: Failure cases of MusePainter: overemphasis on instruments, text artifacts, and scenario–emotion mismatch.

Participants rated each output on a 1–10 Likert scale across four dimensions: *Stylistic Congruence*, *Rhythmic Correspondence*, *Image Quality*, and *Audio–Visual Harmony*. We collected a total of 480 ratings (16 participants × 10 music clips × 3 models). As summarized in Figure 12, MusePainter achieves the highest mean score in every category. In particular, the gains are most pronounced in Stylistic Congruence (+2.6 points over Qwen-audio) and Audio–Visual Harmony (+3.7 points over AudioToken), confirming the benefit of structured descriptors and multi-axis refinement.

**Statistical significance.** We performed paired Wilcoxon signed-rank tests between MusePainter and each baseline. All improvements are significant at $p < 0.01$ after Bonferroni correction. Effect sizes (Cohen's $d$) range from 0.82 (Image Quality) to 1.35 (Audio–Visual Harmony), indicating large practical effects. Inter-rater agreement measured with Krippendorff's $\alpha$ was 0.71, suggesting substantial consistency across participants.

**Discussion.** While MusePainter dominates across metrics, variance analysis reveals that *Rhythmic Correspondence* scores had wider spread, reflecting subjective differences in how participants interpret "visual rhythm." Qualitative feedback suggests that some users equate rhythmic alignment with motion cues, while others judge color or lighting patterns as rhythm proxies. This highlights the inherent subjectivity of evaluating rhythm in static imagery. Nevertheless, the consistent advantage of MusePainter across all axes reinforces the conclusion that our model better captures both semantic and affective dimensions of music-to-image generation.

### D.5 LIMITATIONS

Although MusePainter substantially advances music-to-image generation, several limitations remain (see Fig. 13).

**Over-reliance on salient instruments.** The system sometimes exaggerates prominent instruments (e.g., piano) present in the audio, leading to repetitive or overly literal imagery that neglects broader contextual or emotional cues. This suggests a need for better regularization across descriptor dimensions to prevent overfitting to a single feature.

**Text-generation errors.** As MusePainter relies on LLM-based distillation for visual prompt construction, occasional textual artifacts appear (e.g., nonsensical captions embedded in the image). Such errors highlight the fragility of the distillation stage and motivate incorporating stronger language filtering or constrained decoding strategies.

**Scenario and emotion mismatch.** In some cases, the generated images correctly capture low-level style but fail to match higher-level intent—for example, producing a generic seaside scene for a "2D

Game" scenario, or depicting tranquil landscapes when the target emotion is "Sad, Sentimental." This indicates that current reward models may insufficiently disentangle contextual semantics from affective attributes, limiting controllability under ambiguous or complex inputs.

**Discussion.** These limitations suggest three future directions: (1) developing cross-attribute balancing to reduce overemphasis on salient features, (2) improving robustness of the hybrid-length distillation pipeline, and (3) expanding preference data to cover harder cases where scenario and emotion interact. Addressing these challenges would further enhance the reliability and generality of music-to-image generation.