# OpenReview forum: "Cross-modal Alignment and Human Preference Learning for Fine-grained Music-guided Image Generation"
_ICLR.cc/2026/Conference — ICLR 2026 Conference Withdrawn Submission_

### Official Review · Reviewer_1ZcB · 2025-10-25

**Soundness:** 2
**Presentation:** 2
**Contribution:** 2
**Rating:** 4
**Confidence:** 4

**Summary:**

This paper introduces MusePainter, a framework for music-guided image generation using a three-stage pipeline (FMMD, CMVG, and HCR). It aims to tackle the challenging task of fine-grained music-to-image generation by addressing issues of abstract alignment and subjectivity. However, its methodology and experimental setup are questionable.

**Strengths:**

1. It decomposes the complex task of music-guided image generation into a multi-stage, interpretable pipeline.
2. The paper offers a approach to addressing the core challenges of abstract alignment and subjectivity inherent in music-to-image (M2I) generation.
3. Each of the three stages is broken down with fine-grained, multi-dimensional considerations, aiming for a holistic, multi-axis alignment between the modalities of music and image.

**Weaknesses:**

1. The results only show text-image pairs, omitting the source music. This effectively collapses the evaluation to text-to-image, making it difficult to assess the method's actual music-guidance effectiveness.
2. There is a lack of ablation studies for FMMD.
3. The paper provides insufficient description of the baseline methods, their relevance, and their experimental setups.
4. There are incorrect table citations in the text (e.g., references to non-existent Tables 5, 6, and 8).
5. The subjective evaluation lacks the specific multi-dimensional assessment criteria used in the RLHF training, failing to fully demonstrate the model's contribution to preference alignment.
6. There is significant content overlap between the main text and the appendix. Furthermore, there is a discrepancy in the number of user study participants (16 in the main text vs. 20 in Appendix C.4).

**Questions:**

1. Why are negative samples for RLHF training constructed from the "furthest cluster"? Wouldn't it be more effective to use human-labeled positive/negative pairs generated from multiple runs of the same audio prompt to refine the model's existing capabilities?
2. How were the metrics in Figure 4 calculated? Was this done manually?
3. The RLHF uses a multi-axis reward function. How were the weights for each reward axis determined, and what is the justification?
4. What are the specific configurations of the text-to-music (T2M) models listed in Table 1? What point is this table intended to illustrate, and which T2M model is used in table 1?
5. The BES metric is poorly explained. How are its specific components calculated? How are their weightsdetermined and justified (also for  "Weighted Average Similarity" weights) ?

---

### Official Review · Reviewer_UGNb · 2025-10-28

**Soundness:** 2
**Presentation:** 2
**Contribution:** 2
**Rating:** 2
**Confidence:** 3

**Summary:**

Music-guided image generation with RHLF and fine-grained melodic micro describer. The authors formaize the music-to-image task, point out the difficulty of aligning time-varying music to static images, and proposed a that combines structured music descriptor extraction, cross-modal visual prompt construction, and multi-axis RLHF.

**Strengths:**

1. Proposed a framework that considers multiple aspects of music to guide image generation.
2. Descriptor hierarchy (structural→stylistic→affective) provides interpretable handles for control—useful for creative tooling.
3. Attempts a task-specific evaluation (weighted similarity across emotion/usage/genre/feature) rather than only generic CLIP scores.

**Weaknesses:**

1. No demo website and samples provided, making the work less persuasive.
2. The presentation of the Evaluation Metrics section is messy. The author neither explains the difference between FD and FAD, nor discloses the embedding model for FD and FAD (There is a lack of explanation in KL as well).  Additionally, there aren't any citations after that section.
3. The authors need to introduce all baseline models, at least cite them.
4. 16 participants are not enough for the subjective evaluation, especially for the music-to-image task. Moreover, the definition of the four dimensions is missing.

**Questions:**

1. What is the intention that the authors evaluate music generation models? I can't understand the motivation.
2. I thought the MusePainter is a music-to-image model? I don't see what is Table 1 trying to tell.
3. Implementation section states training on a single server with four A100-40GB and ~150 GPU hours; the Ethics section states a single A100-40GB totaling ~150 hours. Which is it?

---

### Official Review · Reviewer_7x4f · 2025-11-01

**Soundness:** 2
**Presentation:** 2
**Contribution:** 1
**Rating:** 2
**Confidence:** 5

**Summary:**

This paper introduces MusePainter, a framework for fine-grained music-guided image generation. It addresses the challenges of cross-modal alignment between abstract, temporal music and static images by extracting structured music descriptors (structural, stylistic, affective) and mapping them to visual prompts. The pipeline includes three stages: Fine-grained Melodic Micro Describer (FMMD) for music analysis, Coarse Music-Vision Generation (CMVG) for initial synthesis via LLM-distilled prompts and SDXL, and Human-Centric Refinement (HCR) using multi-axis RLHF to optimize emotional consistency, semantic alignment, and creative appeal independently. Contributions include problem formalization, the MusePainter framework, and a benchmark with experiments showing improvements over baselines in metrics like IMSM and BES.

**Strengths:**

The paper demonstrates some originality in integrating structured cross-modal descriptors with a multi-axis preference optimization scheme, which disentangles human feedback axes (emotional, semantic, creative) to avoid reward fusion issues common in RLHF. This approach is reasonably well-argued in the HCR module, with ablation studies (e.g., Table 4) showing its impact on metrics like BES and weighted similarity, providing evidence of targeted improvements.

The framework's emphasis on handling music's subjectivity through hybrid human preferences could offer reusable insights for broader multimodal alignment tasks. Additionally, the presentation is clear, with well-structured figures (e.g., Fig. 2) and reproducible details (e.g., GPU hours, seeds), making it accessible despite the complexity.

**Weaknesses:**

The mapping from music's three descriptor levels (structural, stylistic, aesthetic) to five visual dimensions (color palette, lighting, etc.) relies on an ad-hoc assumption without theoretical justification or empirical validation—e.g., why this specific correspondence is more effective than alternatives, and how it handles variability in musical interpretations. This lacks rigor, as the LLM-based translation in Section 3.3 is described superficially, without analysis of generated patterns (e.g., diversity, potential biases toward stereotypical associations like "sad music to dark colors").

To strengthen this, the authors could conduct qualitative analyses of prompt distributions or ablation on mapping variants.

The music captioning module (FMMD) and related experiments (Tables 1-2) feel tangential and insufficiently justified for the core music-to-image task. Table 1 evaluates downstream text-to-music generation using baselines like MusicControlNet and JASCO, which are temporal-conditioned models not directly comparable. Metrics focus on multimodal alignment (CLAP/CLIP), but omit text-specific ones like ROUGE/BLEU, which are standard for captioning quality and could reveal lexical weaknesses (e.g., paraphrase accuracy). This makes the module's utility unclear—downstream validation is a good idea, but it should be tied more explicitly to image generation improvements.

Human analysis (Fig. 7) lacks depth: with only 16 annotators, no details on their expertise (e.g., musicians?), inter-rater reliability, or statistical tests beyond error bars, it's hard to gauge robustness.

The chosen dimensions (stylistic congruence, rhythmic correspondence, etc.) are not justified—why these over direct measures of the proposed semantic mappings?

Overall, the RLHF contribution, while a strength, feels applied opportunistically to a less meaningful task (music-to-image), where baselines like CLAP already handle coarse alignment; it might better demonstrate value in dynamic tasks like video-to-music or text-to-music, avoiding the "hammer looking for a nail" impression.

**Questions:**

1.  Could you provide justification or empirical evidence (e.g., psychological studies or ablation experiments) for the specific mapping from music's three descriptor levels to five visual dimensions? How does this outperform data-driven or alternative mappings, and what analyses were done on the LLM-generated patterns for diversity and bias?

2. Why include downstream text-to-music evaluation (Table 1) for a music-to-image paper? If it's to validate caption quality, why omit ROUGE/BLEU, and how do the chosen baselines mainly for control modules (MusicControlNet, JASCO) align with your task?

3. For the human study, can you detail annotator backgrounds, inter-rater agreement (e.g., Cohen's kappa), and why the evaluation dimensions were selected? Did you measure the proposed semantic relationships directly, and how might a larger-scale study change results?

4. How does the multi-axis RLHF uniquely advance music-to-image over established tasks like text-to-music, where subjectivity is also high? A response clarifying broader applicability or comparisons could address concerns about task novelty.

---

### Official Review · Reviewer_B4bv · 2025-11-03

**Soundness:** 3
**Presentation:** 3
**Contribution:** 3
**Rating:** 8
**Confidence:** 3

**Summary:**

This paper proposes MusePainter, a music→image generation framework that (i) extracts interpretable, structured musical descriptors (structure/style/affect) to guide visual synthesis and (ii) applies multi-axis preference optimization to separately align emotion, semantic relevance, and creative appeal. The pipeline combines a music describer, a music-to-vision LLM stage, and a diffusion model refined with three reward models via RLHF. The authors also argue that standard cross-modal metrics miss affect/creativity, and introduce a task-specific evaluation (weighted similarity across musical/affective dimensions plus a composite score).

**Strengths:**

- Originality: Neat combination of interpretable musical factors with disentangled preference learning; more principled than pure CLIP/CLAP alignment.
- Quality: End-to-end system with reasonable ablations; human studies go beyond one-number CLIP scores.
- Clarity: Motivation is clear; figures help anchor the pipeline.
- Significance: Addresses a niche but growing cross-modal direction where “affect” matters—not just semantics.

**Weaknesses:**

- RLHF: No head-to-head against a single fused reward under equal budget; a Pareto frontier over emotion/semantics/creativity would be convincing.
- Metric transparency: The composite score and weights need sharper definitions, calibration details, and sensitivity checks to avoid “metric-gaming” concerns.
- Evaluation coupling risk: If the same embedding family appears in mining and evaluation, independence should be demonstrated with alternative feature families.
- Presentation nits: Some cross-refs/labels and human-study protocol details (blinding, agreement, power) could be tightened.

**Questions:**

1. Can you report a single fused-reward baseline (varying weights) with matched compute and show a Pareto frontier across the three axes?
2. Please formalize the composite metric: definitions, weight origins, and a sensitivity analysis.
3. Can you re-run key evaluations with disjoint embedding families to rule out evaluation–mining coupling?
4. Could you detail human-study protocol (blinding, randomization, inter-rater agreement, power)?

---

### Note · Authors · 2025-11-29

I have read and agree with the venue's withdrawal policy on behalf of myself and my co-authors.